# Design and Development of Food Waste Inspired Electrochemical Platform for Various Applications

**Mansi Gandhi** [1,2]

1    Institute of Chemistry, Hebrew University of Jerusalem, Jerusalem 9190401, Israel;
     mansi.gandhi@mail.huji.ac.il
2    School of Innovation in Biodesign, Translational Health Science and Technology Institute,
     Faridabad 121001, India

**Abstract:** Plants have a remarkable position among renewable materials because of their abundance, and nearly thousands of tons are consumed worldwide every day. Most unexploited plants and agricultural waste can be a real potential resource system. With increasing environmental awareness and the growing importance of friendly agricultural waste, crops and fruit waste can be used for efficient conversion into bio-fertilizers, biocarbons, bio-polymers, biosensors and bio-fibers. Global challenges based on limited natural resources and fossil energy reserves simulated keen interest in the development of various electrochemical systems inspired by food and plant scraps, which aid in curbing pollution. The successful adoption of a renewable energy roadmap is dependent on the availability of a cheaper means of storage. In order to cut down the cost of storage units, an improvement on energy storage devices having better stability, power, and energy density with low post-maintenance cost is the vital key. Although food and plant scraps have a huge need for energy storage, it has been extended to various sensing platform fabrications, which are eco-friendly and comparable to organic molecule-based sensors. Current research proclivity has witnessed a huge surge in the development of phyto-chemical-based sensors. The state-of-the-art progresses on the subsequent use of plant-waste systems as nano-engineered electrochemical platforms for numerous environmental science and renewable energy applications. Moreover, the relevant rationale behind the use of waste in a well-developed, sustainable future device is also presented in this review.

**Keywords:** food waste; nano-engineered electrochemical platform; sustainable future; plant scrap inspired; pollution cure



## 1. Introduction

Waste is an unwanted substance generated after the utilization of a valuable part of the matter or the rejected substance, which is considered invaluable, imperfect, and decayed with no possible usage. Often, due to insufficient processing technology or storage units, enormous quantities of agri-produce, especially fruits and vegetables, become wasted [1]. The agricultural waste consists of high carbohydrate content amongst additional multifunctional groups and organic components. Food waste is quite a budding area of research, especially for the generation of highly valuable products, including nano-sized substances. Environmentally friendly nano-systems manufactured from waste obtained from farm waste can serve as an alternative media for clean manufacturing technology in the near future.

Another important research arena is the environmental remediation in extension to the waste systems. Our valuable surroundings are being depleted due to over-exploitation of the environment by human invasions. Environmental contamination is an apprehensive decrease in resources, living species extinction, and ozone layer depletion, and these are some of the most important reasons that led to increased attention on green chemistry. In general, the most crucial principle of green chemistry is the use of renewable resources for

any chemical reaction. The precise use of renewable energy is one of the most effective methods that lead to a significant reduction in environmental pollution. Various production methods of renewable strategies can guarantee the increasing human need for energy in the future. Hence, in this review, we will be agglomerating the major research topics for a sustainable future, i.e., the food waste for sustainable environmental growth and technological advancement (energy, healthcare, sensors, and materials) with the least effect on the environment. Figure 1 is illustrated in this respect, wherein food waste has been extended for various applications such as environmental conservation, energy avenues, nanoparticle formation, and biochemical sensing of human fluids using non-hazardous and simple techniques as a sustainable approach.

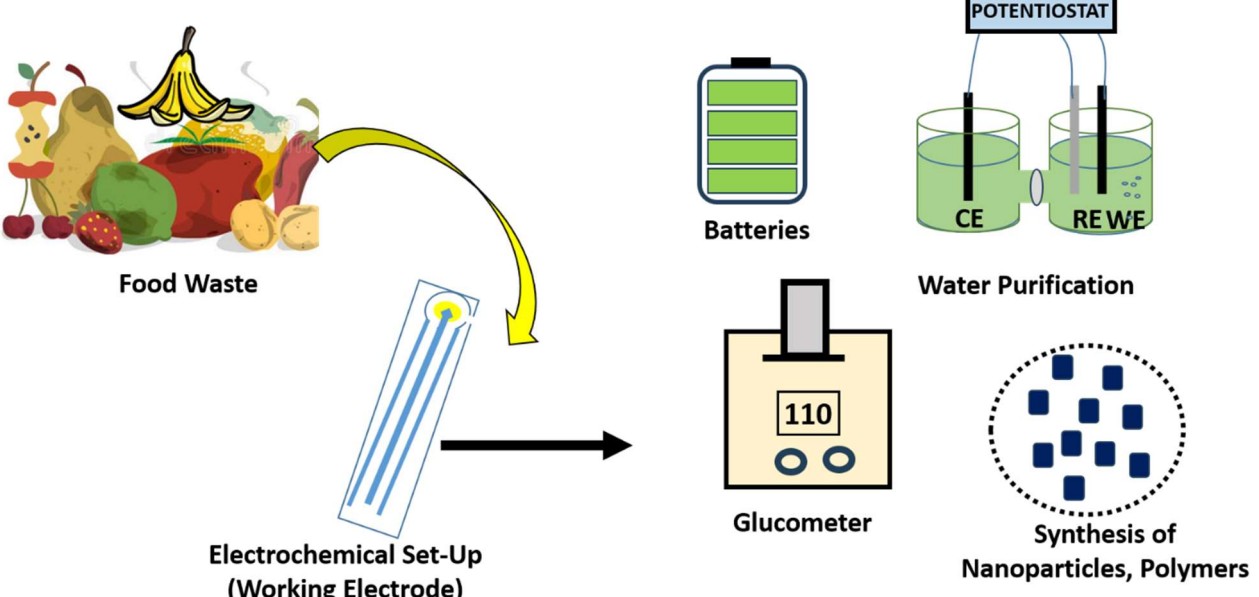

**Figure 1.** Illustration of the plant-generated waste employed for the electrochemical setup opening opportunities for various applications.

Fruits and vegetables are an important constituent of our diets. They help to keep the diseases away. Plants or fruits can be subdivided into edible and non-edible components, where the non-edible ones are generally termed as waste. These wastes are very intriguing materials due to unconventional and unique attributes, which are unveiled in the following review. This can be credited to the presence of antioxidant activity or the free radical scavenging properties of these beneficial dietaries that help reduce the risk of various diseases. They protect us from oxidative stress caused by unfavorable external environments such as drought, injury/accident, virus/micro-organism attack, harmful radiation, etc. This waste consists of important nutrients which could be recycled into useful materials that could be further reused for different purposes. This recycling of biowaste could be a potent bio-stimulant fertilizer, consisting of peels, leaves, pseudo stems, and stalks, for various foods and non-food applications. They can be explored as an alternative source for macro or micronutrients, livestock feed, natural fibers, and sources of natural bioactive compounds. Various renewable plant and vegetable waste-generated bio-waste or biomasses can be employed in the production of bio-carbons that exhibit exceptional properties. These wastes include coffee waste, vegetable peels (potato [2], mango [3], garlic [4], corn stover, grapefruit [5], banana [6], onion [7], etc.), leaf litter [8], almond shells [9], willow catkin [10], chitosan, rice straw [11], etc. This waste, apart from being just a waste or environmental pollutant, can be a remarkable aspect for unprecedented use as an abundant resource with high saccharide content and also be a bio-based porous carbon [12]. These eco-friendly carbon-based materials/compounds have exceptional attributes owing to their superior electrochemical properties of high surface area, and have the potential for use in

high-performance dye-sensitized solar cells (DssC) and supercapacitors. Their application can enhance high power conversion efficiency, controllable morphology (honeycomb [12], graphitic [13], crystalline [4], powdered, and porous carbon foams [14]), and remarkable cycling stability. Even a paucity of employing these biocarbon-based materials as counter electrodes can be applied to DssCs. Various reports account for the presence of antioxidant activity in fruits that are generally more concentrated in peels than in the flesh, representing a major fruit protection barrier [15,16]. The most promising example is the apple, wherein the concentration of ascorbate or vitamin C is six times more concentrated in the peel compared to pulp. In addition, the antioxidant concentration in the peel is very high, a quantitative indicator of the status of the antioxidant-based defense system of the fruit.

The antioxidant capacity of fruits is usually determined by synthetic radical capture and spectroscopic monitoring (UV-Vis). A number of high-pressure liquid chromatographic technique (HPLC) methods have been developed for flavonoid and ascorbic acid determination in fruit peels. But, these methods are time-consuming and costly to operate. Hence, electrochemical detection has widespread utilization with minimal sample input and high selectivity for electroactive species [5]. These techniques are categorized by unfriendly reagents to the environment, pre-treatment sampling, and long reaction times, paving the way for the electrochemical index (EI) concept. The EI is based on anodic and cathodic peak current independently, wherein the current intensity is directly proportional ($\alpha$) to the electron transfer rate, which is directly proportional ($\alpha$) to the electroactive species present.

The presence of various flavonoids, quercetin, natural antioxidants, and polyphenolics is a huge part of fruit peels. The accumulation of potential health benefits has been discovered in anti-inflammatory, anti-carcinogenic, anti-diabetic, and anti-allergic properties. The waste generated from the juice industry is a resource that causes environmental pollution. Moreover, this peel, when added to the soil, helps in the better germination of crops and enhances soil quality [17]. In this review, we took a holistic approach that compiles the use of herbivore product function and various applications of these moieties in biomaterial engineering research, biofuel, and artificial biosensors for a facile, rapid and generic biosensor platform. Meanwhile, their usage in corrosion inhibition, oxidation-reduction reactions, and preparation of nanoparticles are usually not given many credentials.

## 2. Different Varieties of Food and Plant Generation Waste and Their Significance

Different plant wastes such as aloe skin [12], mangosteen, orange scrap [18], sapota waste [19], and papaya peels [20] are available in nature. These plant-generated waste materials are an important source of soluble and insoluble fibers [21] to fight cholesterol [22], fight antioxidants [23], and protect the eyesight [24]; other uses include their viability as a meat tenderizer [25], as a teeth whitener [26], as a polishing agent [27], in skin and beauty regimes [23] and even for healing psoriasis [28], and as a garden booster to deter crawling pests [29]. These uses can be attributed to the presence of different metal ions such as manganese, sodium, calcium, zinc, phosphorous, magnesium, potassium, and folic acid [21,30]. These are essential for promoting general plant vigor, build-up, and resistance to pests and diseases, which are necessary to help fruit grow and are involved in regulating their defense mechanism. Some important routine facts that involve the use of banana skin for the production of bioethanol using the yeast *Saccharomyces cerevisiae* were reported by Gebregers et al. in 2016 [31]. Similarly, in 2020, Bouziz et al. reported on bioethanol production using date seed cellulosic fraction [32]. Even this has significant effects on various biological aspects of plants, such as seed germination, growth promoters, etc.

## 3. Waste-Inspired Electrochemical Applications

Plant waste, vegetable scrap, and fruit skin are a source of environmentally friendly as well as cost-effective sorbent systems for the removal of many heavy metals like $Cu^{2+}$, $Pb^{2+}$, and methylene blue from aqueous systems [33]. The elimination of contaminants present in the water bodies due to anthropogenic activities is one global issue [34]. The Environmental Protection Agency statistics of 2012 demonstrated that ~38 million liters of

acid sulfate copper were spilled into the Sorona and Bacanuchi Rivers, thus affecting the water supplied to about 25 thousand residents. Hence, necessary alternatives for the remediation of metal ions in polluted or wastewater cleaning are required. To solve this problem, functionalized carbon generated using vegetable and fruit waste can be a double benefit, i.e., it can be advantageous as a cheap bio-sorbent material for the removal of fluoride ions and copper ions from groundwater. It is a well-established de-fluorinating adsorbent and almond shell-based electrochemical platform [9,35]. Further, they have been extended as anti-counterfeiting sensors and ion probes [36–39] for various applications. A non-invasive electrochemical antioxidant activity measurement technique has been reported to use soft carbon microelectrodes for apple skin [40], which helps in counterbalancing oxidative stress when using the scanning electrochemical microscopic technique. In comparison, traditional techniques such as reflectance [41] and chlorophyll fluorescence spectroscopy [42,43] affect the concentration as they are UV-sensitive and invasive protocols. Therefore, the development of simple, unique, and non-destructive sensors to rapidly analyze with high spatial resolution at a low cost with an onsite antioxidant defense system is attractive for cultivators and farmers.

## 4. Bio-Generated Carbon Template/Systems for Energy Storage

Carbons generated after waste combustion/oxidation/pyrolysis, i.e., the black residue left after biowaste, are thermochemically converted. Thus, carbon sequestration through biochar can be a very good base for soil, acting as fertilizer, and sequestering carbon has a mean residence time of about 2000 years [44–47].

Few research organizations have produced biofuels and carbon chemicals (such as $CH_4$, ethanol, hydrogen-rich syngas, and benzene family derivatives). A recent account involves the open and closed burning of rice husks, giving different properties of the synthesis process. Thus, the processing and synthesis method decides the customized nature of the carbon type. These products can be very suitable based on distinct properties obtained under various circumstances. The use of rice husks can lead to carbon nanomaterials like biochar, graphene, graphene oxide, carbon nanotubes, and so on. Nanaji et al. involved jute sticks in the synthesis of a graphene-like nanoporous carbon [48] and studied its electrochemical properties. Another report accounts for the plasma synthesis of graphene from mango peels [45].

The reliance of energy utilization on fossil fuels is predicted mainly through variables like world economics and ecology, i.e., depleting resources and escalating environmental concerns. In this respect, nanomaterials play a key role, especially in porous carbon-based systems. These systems have been a major hotspot in the research arena due to their potential applications (variant characteristics like abundant availability, high surface-to-volume ratio, porous structure, active sites, economic and long cycle life, etc.). The involvement of plant/agricultural waste in the production of energy storage or conversion devices with high-power applications is a novel concept. A hybrid power source comprising a supercapacitor in parallel configuration with a battery is proposed for short-duration pulse devices with a high specific power. The superior power density is rooted in their charge storage mechanism. The primary type of supercapacitor accounts for the electrical double-layer capacitance, where charges are stored owing to their ions getting adsorbed at the electrode/electrolyte interface. The second type is pseudo-capacitance reactions, wherein surface redox reactions account for fast ion insertion and extrusion, triggering no transition in their phases. But, the major drawback of these graphene-based carbon nanotubes is in their bulk production and practical implications, which are affected due to the lack of a simple synthesis process or cost-effective raw material.

On the contrary, biomass waste is the best precursor due to its low cost, high abundance in waste disposal, and carbon content. A few examples of such systems are cherry stones, bamboo, candlenut shells, pecan shells, vine shoots, lotus stalks, toddy palms, chicken feathers, etc. Extensive studies have worked on electrochemical energy storage in the last few years. Generally, researchers have been working on the incorporation of superior

features. Herein, the review article talks about energy systems developed using plant product skin-inspired carbon systems. The involvement of renewable sources replacing chemicals not only stops environmental contamination but also helps to decrease the production cost. Nearly one billion metric tons of fruit scraps are annually produced, indicating their relevance in human lives. This zero-cost scrap can be employed as a substitute for hazardous chemicals, as tabulated in Table 1. The use of paste waste has been studied as a corresponding material for solar cell fabrication. The various attributes of solar cell comparative analysis based on their open-circuit potential and power conversion efficiency (i.e., input vs. output) have been tabulated accordingly.

**Table 1.** Tabulation comparison of various bio-source-derived carbon as a counter electrode for dye-sensitized solar cells (DssCs). ($V_{OC}$—open circuit potential; $J_{SC}$—short circuit current; **FF**—fill factor; and **PCE %**—power conversion efficiency).

| S. No. | Counter Electrode Derived from BioSource | $J_{SC}$ | $V_{OC}$ | FF | PCE % | Ref |
|--------|-------------------------------------------|----------|----------|------|-------|------|
| 1. | Aloe Peel | 14.15 | 720 | 0.68 | 6.92 | [12] |
| 2. | Coffee Waste | 15.09 | 760 | 0.72 | 8.32 | [49] |
| 3. | Pine Cone Flowers | 13.51 | 710 | 0.51 | 4.98 | [50] |
| 4. | Pumpkin Stem | 3.84 | 611 | 0.47 | 2.79 | [51] |
| 5. | Sunflower Stalk | 15.20 | 670 | 0.64 | 6.56 | [52] |

Highly porous activated carbon can be prepared for the development of symmetric supercapacitor devices. Various supercapacitor anodes include coffee waste [53] and rice husk [54], which consist of a high degree of electrochemical stability, robustness of electrode material, and reversibility. While citrus fruit-derived carbon shows well-defined electrochemical energy storage reported by various reviewers [55].

The carbonization of coffee waste with various composites is used for electrochemical energy storage, i.e., nitrogen-doped carbon coffee waste–$ZnCl_2$ [49] via an economically viable method. At the same time, these systems exhibit high yields with more active sites and channels for migration and adsorption of electrolyte ions, resulting in superior specific capacitance values and high energy density features. Figure 2 accounts for the involvement of easily available waste like papaya fruit peels, cow dung, and jackfruit peels for the generation of various carbon types. Taer et al. reported seven different types of activated carbon electrodes using peel waste for various supercapacitors having promising features of high surface area, and the cost to gain the material is zero [56]. Other distinct varieties of electrodes for supercapacitor-based devices were made using various plant-scrap waste, for example, 'corncob [57], coconut shell fibers [58], sugar cane bagasse [59], oil palm empty fruit bunches [60], cassava scrap [61]', exhibiting monolithic systems without adding any adhesive material. The electrochemical properties of carbon peel wastes are found using the cyclic voltammetric technique, wherein the voltage and current density relationship for the samples with various carbonization conditions are measured. The shape of the *I-V* curve portraying a square shows almost an ideal shape with higher current density at relatively low voltage. These data exhibit the process of ion distribution into the meso and micropores. Lv et al. reported a self-template synthesis of a hierarchical porous carbon foam-inspired supercapacitor electrode that could accommodate zinc ions via adsorption, leading to zinc complexes [14].

The scope of nitrogen-doped carbon nanosheets has been shown using easily available garlic peels as a precursor for the synthesis of a graphene analog synthesized via a cost-effective/economic method, as shown in Figure 2C's reprint from [13]. The functionalized systems exhibited excellent electrochemical properties, i.e., electrochemical performance in Na/Li cell types. The interesting property of the pores reduces the ion diffusion pathway and helps in easy electrolyte access, thereby increasing the ion uptake. The as-prepared systems had a defined capacity retention comparable to the reported anode energy storage

materials with a high energy density and long cycle life [13]. Such analogous systems are accounted for in Table 2.

**Table 2.** Accounts for various renewable plant-based energy devices with their characteristics.

| S. No. | Name of Waste | Specific Surface Area | Capacitance Retention | Current Density | Reference |
|---|---|---|---|---|---|
| 1. | Aloe Peel | 1286 m$^2$ g$^{-1}$ | 91% | 30 A g$^{-1}$ | [12] |
| 2. | Banana Peel | 1362 m$^2$ g$^{-1}$ | 100% | 5 A g$^{-1}$ | [6] |
| 3. | Cassava Peel | 398 m$^2$ g$^{-1}$ | 66% | - | [61] |
| 4. | Coconut Husk | 1000 m$^2$ g$^{-1}$ | - | 0.05 A g$^{-1}$ | [58] |
| 5. | Garlic Peel | 1710 m$^2$ g$^{-1}$ | 95% | 1 A g$^{-1}$ | [13] |
| 6. | Melo Fruit Peel | 721 m$^2$ g$^{-1}$ | 91% | 1 A g$^{-1}$ | [62] |
| 7. | Pomelo Peel | 807 m$^2$ g$^{-1}$ | 100% | 5 A g$^{-1}$ | [63] |
| 8. | Rice Husk | 1768 m$^2$ g$^{-1}$ | 95% | 0.05 A g$^{-1}$ | [54] |
| 9. | Shaddok Peel | 2475 m$^2$ g$^{-1}$ | - | 0.5 A g$^{-1}$ | [64] |
| 10. | Willow Catkin | 645 | 92% | 0.1 A g$^{-1}$ | [10] |

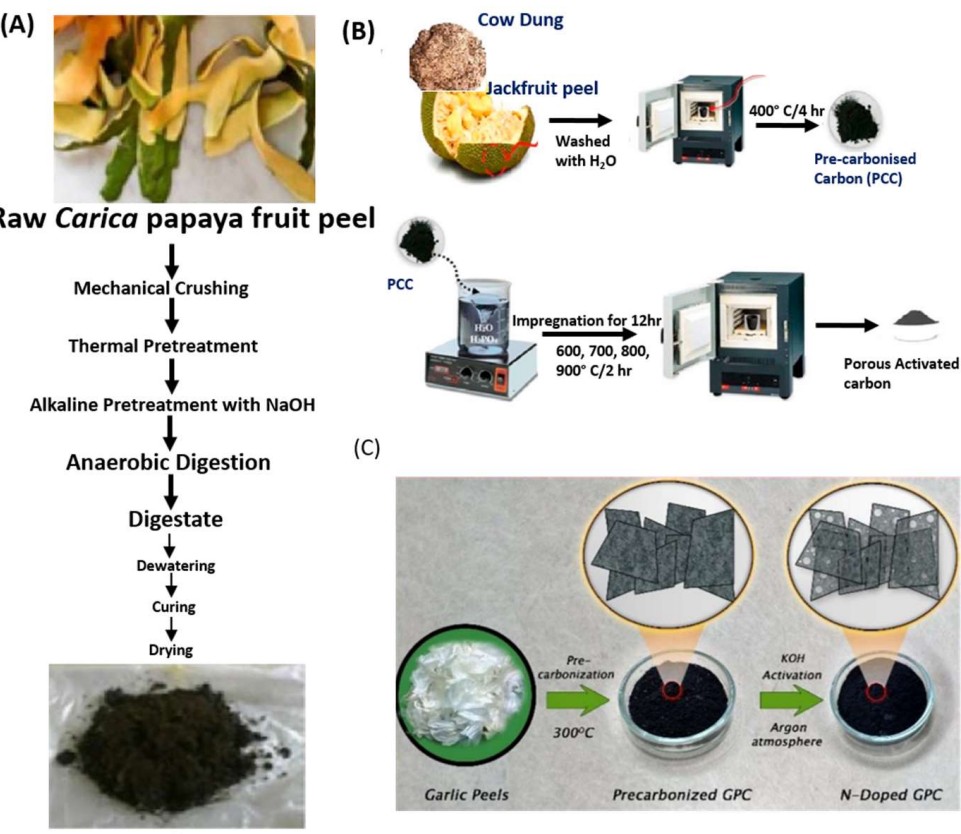

**Figure 2.** (**A**) Diagrammatic representation of the papaya peel-inspired carbon reprinted with copyright permission [20]. (**B**) Jackfruit peel and cow dung-based pre-carbonized carbon preparation reprinted with copyright permission [65]. (**C**) Garlic peel-based graphitic mesoporous carbon formation and its functionalization, reprinted with copyright permission [13].

## 5. Mapping of Biomass Generation

The procedure for these systems includes a one-step pyrolysis and hydrothermal carbonization or two-step chemical activation carbonization. The one-step carbonization procedure leads to high decomposition rates but low carbon yields and heterogeneous carbon pores. On the contrary, hydrothermal carbonization leads to increased yield, although some constraints for the low specific surface area have been reported by Liu et al. and Hu et al. [66,67]. The as-prepared carbon functionalized system was prepared by employing a

straightforward hydrothermal carbonization combining alkali activation methods in order to increase the yield of carbon with rich porosity attributes. This is based on dehydration and decarboxylation for the hydrothermal carbonization stage, followed by the reaction of alkali activator and carbon that occurs in the pyrolytic activation stage. Biochar is widely recognized as a multifunctional material that can develop high-performance carbon materials for both energy and environmental applications, which remains a challenge. These methods are displayed in Figure 3, reprinted from [57,58]. Figure 3 involves the fabrication of a supercapacitor using a coconut husk and corn cob, which can be a new approach towards the use of biodegradable waste systems as an alternative to synthetic organic molecules that require expertise with sophisticated instrumentation and chemical systems.

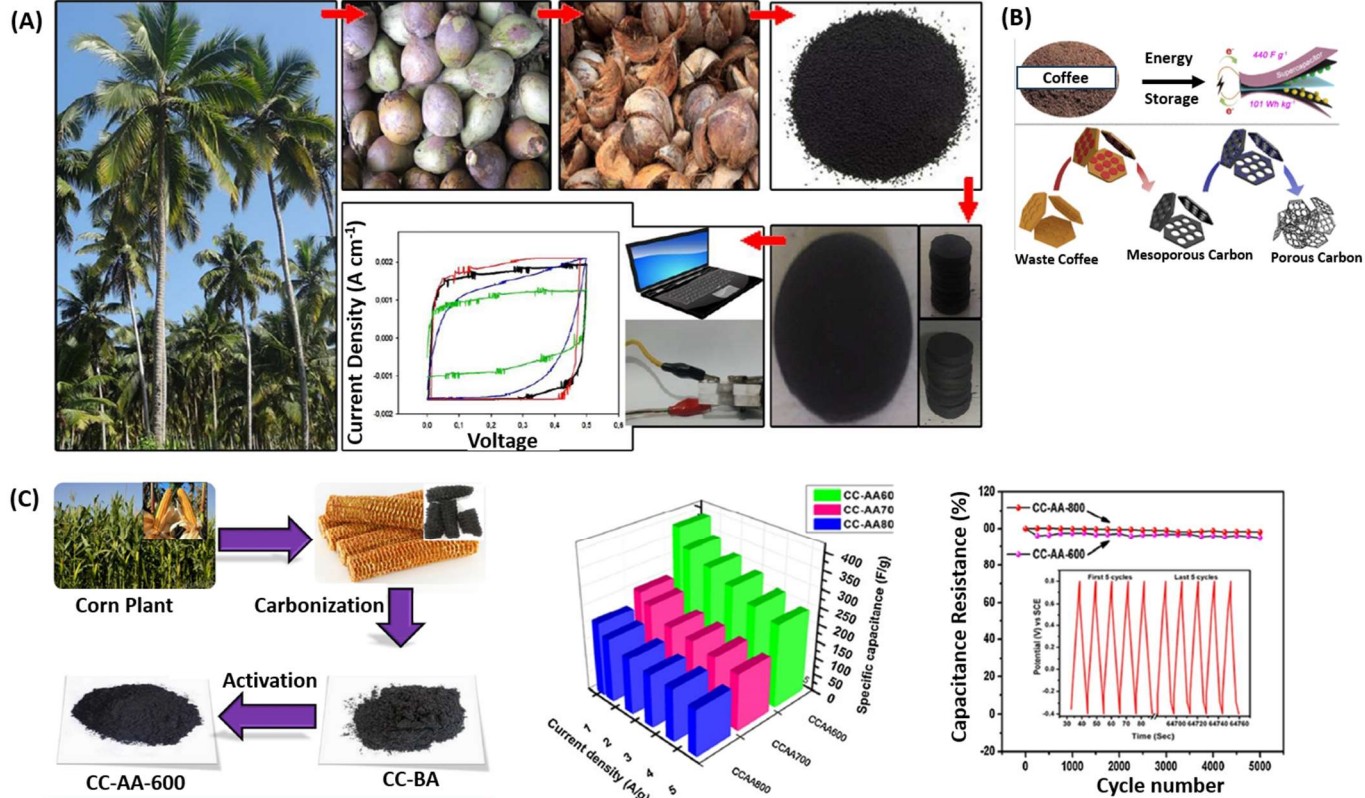

**Figure 3.** (**A**) Diagrammatic representation for synthesis of activated carbon using coconut husks and the subsequent use in supercapacitors, reprinted with copyright permission [58]. (**B**) High−yield waste coffee ground-based mesoporous carbon for supercapacitor application, reprinted with copyright permission [53]. (**C**) Porous carbon-inspired supercapacitor for aqueous and non-aqueous electrolyte system based on corn cob waste-inspired system, reprinted with copyright permission [57].

Aloe peel-derived 3D honeycomb-like porous carbon was developed using hydrothermal carbonization in an interesting report. These carbon-based systems exhibit a large surface area that helps in building superior properties for dye-sensitized solar cells (DssCs) [12]. Furthermore, the electrochemical–catalytic performance is based on the two characteristics of increase in current density $i_\mathrm{p}$ and peak-to-peak separation $\Delta E_\mathrm{p}$, which corresponds to the charge transfer process at the electrolyte–CE interface and Nernst ion diffusion in the electrolyte solution followed by an $I_3^-$ reduction comparison.

After the basic preparation of carbon-based systems, their activation is important, which is procured by various physical and chemical activation processes. The physical activation process involves the removal of a large quantity of internal carbon mass with well-built carbon. At the same time, the chemical activation involves methods such as dehydrating agents that affect the pyrolytic decomposition, resulting in the materialization of tar with increased carbon yield [68–70]. Table 3 is a comparative table for a Pomelo-

based carbon system used for supercapacitor application, showing different properties and characteristics due to the method of preparation.

**Table 3.** Accounts for various Pomelo fruit peel-generated carbon systems, their constraints such as potential window of operation, specific surface area, current density, and cycle stability, and its deficiency and specific capacitance.

| Fruit Peel | Specific Surface Area ($m^2 \ g^{-1}$) | Potential Window | Specific Capacitance ($F \ g^{-1}$) | Current Density ($A \ g^{-1}$) | Stability Cycles (number) | Cycle Deficiency% | Ref |
|---|---|---|---|---|---|---|---|
| | 999 | −1 V to 0 V | 338 | 1 | 5000 | 4 | [71] |
| | 38.44 | −1 V to 0 V | 222.6 | 0.5 | 5000 | 3 | [72] |
| Pomelo | 2167 | 0 V to 3 V | 1115 | 0.2 | 2000 | 9 | [73] |
| | 830 | −1 V to 0 V | 321.7 | 1 | 6000 | 6.6 | [64] |
| | 1265 | −0.1 V to 0.9 V | 550 | 0.2 | 10000 | 6.3 | [74] |

Furthermore, the pores cause an increase in surface area and better volume of the activated biomass carbon. These pores enhance the contact between the electrode–electrolyte interface and shorten the ion diffusion pathways, delivering the minimum diffusive resistance. Qian et al. explained the importance of temperature for the development of pores, as higher heating temperature leads to the collapse of pores. Thus, higher surface area and narrower pore size distribution are advantageous for charge transfer, as reported by Qian et al. in 2014 [75]. The activation process was carried out using phosphoric acid treatment followed by either calcination or roasting.

## 6. Phyto-Nutrient-Based Sensing Platform

The detection of biologically active molecules is of critical importance from a biomedical, environmental, and security point of view. Such detection can be carried out by bioanalytical protocols. Sensors consist of two major parts, namely, the receptor and the transducer element. The receptor is generally an organic or inorganic material having specific interaction with one analyte or a group of analytes. In the case of a biosensor, the receptor/recognition element is a biomolecule. In comparison, the transducer element is the one that converts the chemical information to a measurable signal.

Biosensing is of paramount importance for improving the quality of human life. Biosensors are of great help in detecting a wide range of compounds sensitively and selectively, with applications in security, healthcare for point-of-care prototypes, and environmental safety [76–78]. Figure 4 shows a grapefruit-based electrochemical biosensor for copper detection, as reported by Romero et al. [79].

The use of a bio-template-based carbon electrode system is quite efficient and economical and has vivid environmental applications such as capturing carbon dioxide, sensing platforms, and a lot more [80]. The use of agro-industrial waste helps in obtaining a synergy for the development of sensing signals/response output. An interesting mobile device was reported by Romero-Caro and co-workers for the detection of copper ions using a carbon paste electrode prepared using grapefruit peels as a bio-template platform involving a differential pulse voltammetric approach. This electro-analytical platform offers the advantages of high sensitivity and easy operation, with portability options. To gauge the improvement in performance, the as-prepared platform of the assay was validated.

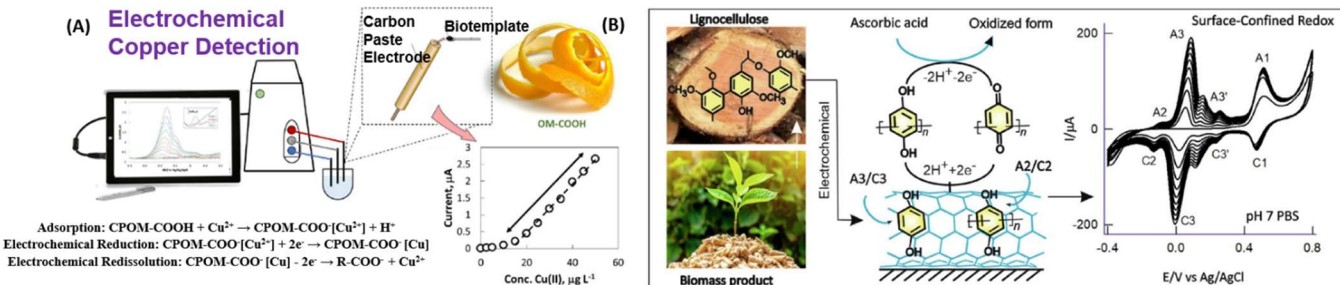

**Figure 4.** (**A**) Reprint of the grapefruit scrap (OM−CCOH)-based system for electrochemical copper detection with copyright permission [79]. (**B**) Lignocellulose biomass product syringaldehyde for electrochemical-based fuel cell applications, reprinted with copyright permission [81].

## 7. Advantages of These Eco-Friendly Alternatives

Agricultural waste and byproducts present problems for disposal and the degrading odd odor disliked by all. These wastes do not have much utility at present and have been classified as pollutants to the environment. Decreasing the amount of waste pollution from the surrounding is the major advantage of these systems.

The skin, peel, or scraps can be a potent eco-friendly catalyst system for biodiesel synthesis, as reported by Etim et al. in 2018 [82]. These systems have also increased the durability of a new generation for the evolution of electronic essentials.

The prominent use of renewable and facile materials in various chemical industries is not just due to decreased cost, but because it also leads to a decrease in environmental pollution and the waste becoming harmless. Thus, the scope for applications in material science for removing heavy metals and organic compounds from aqueous solutions and oxygen reduction reaction catalysts for reduction reactions is widened, as they are a sustainable, facile, and economical source with which to fabricate appropriate resources.

Waste-generated carbon-based systems have been explored in distinct domains and have shown a well-defined usage in the arenas of sensor and memory device fabrication [36].

## 8. Downside of Using These Systems

The presence of metal impurities in different peels is not always conductive enough, hence leading to the formation of disordered pore structures; therefore, endogenous minerals which could have an inhibitory effect should be kept in mind before initiating any applications and carbonization processes [64].

Usually, many metal oxides and salts consisting of ions such as K, Na, Ca, Mg, etc., exist inside the biomass and contribute to the impact on biomass pyrolysis. Hence, they help in the formation of coke and gas products, which are a pollutant, while it reduces the stability of the carbon skeleton [82].

The presence of metal ions in biomass-derived carbon can interfere with its application and can sometimes limit its potential functions, as in the study by Han and colleagues in 2021, where the garlic peel was demineralized [4].

Another constraint is the choice of carbonization temperature, as carbon is prone to collapse at higher temperatures and cannot form subsequent pores, resulting in a rapid decrease in the specific surface area of pores [67].

The most challenging part lies in the designing of specific and economical materials involving electrodes that help to improve accuracy and precision and ensure selectivity.

## 9. Conclusions and Future Prospects

The peel waste-inspired electrochemical platforms are abundant, renewable, inexpensive, and environmentally benign in comparison with artificial templates and precursors. The results of the review article provide a new path and blueprint for designing much more

stable electrochemical capacitors using an environmental synthesis route and less toxic electrode and electrolyte media. In addition, peels and scraps are low-cost and environmentally friendly, and appropriate electrochemical applications are crucial and economically justified. Few of the other eco-friendly green methods for the synthesis of mesoporous carbon having good electrical conductivity, mechanical stability, and larger surface area, economic output, and density can be carried out using precursors such as petroleum, coke peanut shells, juglone, indanthrone blue dye, etc.

Nevertheless, in order to consider this practice as a green and sustainable process, some significant points should be kept into consideration about the reuse of waste products, recycling of exhausted systems, and use of non-synthetic and environmental substitutes. Furthermore, the suitability and feasibility of utilizing these byproducts are future prospects, and directing them as sources of natural bioactive compounds is quite a challenging task. Moreover, the byproducts obtained from the agriculture and food processing industries can be explored in the near future as promising platforms. This is of pivotal importance in maintaining environmental harmony.

**Funding:** This research received no external funding.

**Acknowledgments:** The author would like to acknowledge the Hebrew University of Jerusalem for the completion of her postdoc.

**Conflicts of Interest:** The authors declare no conflict of interest.

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
