# Peer review of "Design and Development of Food Waste Inspired Electrochemical Platform for Various Applications"

_2673-3293, doi:10.3390/electrochem4030026_

Round 1
Reviewer 1 Report
The review is concise. Yet, the provided figures taken from other publications (with permission) are unclear. The graphs are blurred. So, replacing high-resolution graphics is suggested.
Author Response
Ref: electrochem-2473850
Title: Design and Development of Food Waste Inspired Electrochemical Platform for Various Applications
- Reply to the Reviewer #1’s comments:
The review is concise. Yet, the provided figures taken from other publications (with permission) are unclear. The graphs are blurred. So, replacing high-resolution graphics is suggested.
Reply: Thank you for the feedback. We have suitably incorporated the suggestions (Figure 2, Figure 3, and Figure 4) and added new high-quality Figure 1 in the revised manuscript, making it suitable for publication.
Reviewer 2 Report
The review article of M. Gandhi deals with a current and important area, but I recommend its publication only after proper revision. Irrelevant parts should be deleted and the currently listed applications should be explained in more detail.
My other suggestions are as follows:
First two sentences of Abstract are unnecessary.
Introduction
The purpose of the introduction in a review article is to outline the problem and show what areas the article will cover. This is missing in present paper, the chapter is not well structured. It talks about plant waste, but also about uses and antioxidant measurement methods, without a logical system. E.g. The last two sentences of 3. paragraph rather belongs to 5. paragraph.
I suggest rethinking and editing the entire introduction
Chapter 2.
The title is meaningless. I think the topic of this chapter would have been the generation of plant waste, but then let the chapter be about that. There is no need to list the uses of the components found in waste here (especially if they are not electrochemical, starting with insoluble fibers and ending with bioethanol).
Chapter 3.
Why do you consider a vegetable fiber-based sorbent system as an electrochemical application? ( for adsorption of methylene blue, metal ions or fluorine)
Ref 35 is rather 34.
Ref. 40 describes a non invasive measurement method!
„While traditional techniques such as reflectance[41] and chlorophyll fluorescence spectroscopy[42, 43] affect the concentration as they are UV sensitive and invasive protocols.”
While traditional techniques such as reflectance[41] and chlorophyll fluorescence spectroscopy[42, 43]are invasive protocols and affect the concentration of the UV sensitive analytes, like antioxidants.
Chapter 4.
The abbreviations of Table 1 must be given in a footnote or in the text.
It is necessary to interpret the data of Table 1. in the text. Similarly, Fig.1A and Fig. 1B should be mention in the text, and data of Table 2, as well.
Chapter 5.
Title is innacurate. Chapter deals with carbonization process for production of biomass based activated carbon.
Chapter 6.
Parts A and B of Figure 3 should be placed one below the other. Please, explain how the biosensor and the fuel cell works.
The quality of the figures is not adequate, they are too crowded.
The ref list is inaccurate, verification is required, e.g. 19.23.29. 44. 76.78.
Author Response
Ref: electrochem-2473850
Title: Design and Development of Food Waste Inspired Electrochemical Platform for Various Applications
- Reply to the Reviewer #2’s comments:
The review article of M. Gandhi deals with a current and important area, but I recommend its publication only after proper revision. Irrelevant parts should be deleted and the currently listed applications should be explained in more detail. My other suggestions are as follows:
Reply: Thank you for the critical feedback. We have suitably incorporated all the suggestions and corrections in the revised manuscript, making it suitable for publication.
- First two sentences of Abstract are unnecessary.
Reply: Thanks for your critical comment. We agree with your point and have suitable edited in the revised version of manuscript.
- The purpose of the introduction in a review article is to outline the problem and show what areas the article will cover. This is missing in present paper, the chapter is not well structured. It talks about plant waste, but also about uses and antioxidant measurement methods, without a logical system. E.g. The last two sentences of 3. paragraph rather belongs to 5. paragraph. I suggest rethinking and editing the entire introduction.
Reply: Thanks for your feedback. the critical comment. The last two lines of paragraph 3 of Introduction are “The most promising example is the apple wherein; the concentration of ascorbate or vitamin c is six times more concentrated in the peel compared to pulp. In addition, the antioxidant concentration in the peel is very high, a quantitative indicator for the status for antioxidant-based defense system of the fruit.” While the last two lines of para 5 are “In this review, we have taken a holistic approach which compile the use of herbivore-product function and also various applications of these moieties in biomaterial engineering research, biofuel, artificial biosensors for a facile, rapid and generic biosensor platform. Meanwhile their usage for corrosion inhibition, oxidation-reduction reactions and preparation of nanoparticles are usually not given much credentials.” Both paragraphs are very different from each other.
1st paragraph of introduction talks about the generation of agri-waste which can be a budding area of research hotspot.
2nd paragraph talks about the environmental remediation of these waste systems. Utilizising these waste systems as a crucial part of green chemistry for various different application domains.
3rd paragraph talks about the different varieties of plant waste products peels, leaves, pseudo stem, stalk, etc. and their specific usage into application like bio-charred carbon formation, energy storage, material modification (solar cells), their antioxidant capacity and their nutritive value in terms of saccharide content, fiber, micro and macro nutrient, etc.
4th paragraph talks about electrochemical techniques used for the antioxidant activity along with quantitative and qualitative determination of these plants using the concept of electrochemical indexing.
5th paragraph talks about the closing of introduction with a cumulative summing up about the plant waste, their properties and their uses. Furthermore, their amalgamation for the development of engineering research, biofuel, artificial biosensors for a facile, rapid and generic biosensor platform can be potential arenas of research.
- Chapter 2. The title is meaningless. I think the topic of this chapter would have been the generation of plant waste, but then let the chapter be about that. There is no need to list the uses of the components found in waste here (especially if they are not electrochemical, starting with insoluble fibers and ending with bioethanol).
Reply: Thanks for your comment. We agree with your point. Hence, as suggested by the reviewer the Section 2 has been retitled as “Different Varieties of Food and Plant Generation Waste and their Significance”. The paragraph talks about the various plant waste varieties available in nature and their respective usage in various domains such as fighting cholesterol, garden booster, resistance to pest and diseases, heals psoriasis, teeth whitener, production of bioethanol, etc. This paragraph is a culmination of all the various applications other than electrochemical for a brief background to the readers.
- Chapter 3. Why do you consider a vegetable fibre-based sorbent system as an electrochemical application? (For adsorption of methylene blue, metal ions or fluorine). Ref 35 is rather 34.
Reply: Thanks for your comment. We have updated the reference list in the revised version of manuscript. There are examples taken in support to the title of the article as “Food waste inspired Electrochemical Platform”. Hence, the example are based on this.
Ref 34: Removal of methylene blue dye from aqueous solutions by adsorption using yellow passion fruit peel as adsorbent
Ref 35: Coupled Adsorption and Electrochemical Process for Copper Recovery from Wastewater Using Grapefruit Peel.
- 40 describes a non-invasive measurement method!
While traditional techniques such as reflectance [41] and chlorophyll fluorescence spectroscopy [42, 43] affect the concentration as they are UV sensitive and invasive protocols.”
While traditional techniques such as reflectance [41] and chlorophyll fluorescence spectroscopy [42, 43] are invasive protocols and affect the concentration of the UV sensitive analytes, like antioxidants.
Reply: Thanks for your comment. But the ref corresponds to non-invasive technique only. The statement from the manuscript is “A non-invasive electrochemical antioxidant activity has been reported using soft carbon microelectrodes for apple skin[41]”.
Ref is Mapping the antioxidant activity of apple peels with soft probe scanning electrochemical microscopy
While the other references for traditional techniques of reflectance and fluorescence spectroscopy is quite correct.
- Chapter 4. The abbreviations of Table 1 must be given in a footnote or in the text.
Reply: Thanks for your comment. We have suitably added it in the revised version of manuscript.
It is necessary to interpret the data of Table 1. in the text. Similarly, Fig.1A and Fig. 1B should be mention in the text, and data of Table 2, as well.
Reply: Thanks for your comment. We have suitably incorporated in the revised version of manuscript (Table 1). The Figure and Tables have been explained in the revised version of manuscript as suggested by the reviewer.
- Chapter 5. Title is inaccurate. Chapter deals with carbonization process for production of biomass based activated carbon.
Reply: Thanks for your comment. The section 5 deals with the mapping of biomass through the plant and peel waste. Since they are the part of organic systems, the ultimate end product of organic decomposition is carbon. But this Biochar carbon is result of pyrolysis and carbonization can be very advantageous and has many applications involved, if tailored during decomposition E.g., Aloe–peel derived 3D honey comb like porous carbon have been developed using hydrothermal carbonization which has been extended for dye sensitized solar cells. Hence, we feel the title is quite considerable for this topic.
- Chapter 6. Parts A and B of Figure 3 should be placed one below the other. Please, explain how the biosensor and the fuel cell works.
The quality of the figures is not adequate, they are too crowded.
Reply: Thanks for your comment. We have updated the Figures suitably.
We regret to say that explanation of a biosensor or a fuel cell would deviate this paper from the topic. Hence, we could not accommodate into the manuscript revision.
The ref list is inaccurate, verification is required, e.g. 19.23.29. 44. 76.78.
Reply: Thanks for your comment.
Ref 19 is “Electrochemical Investigation of Manilkara zapota fruit Peel Extract on Mild Steel in Acid Medium”. The ref has been used in the statement regarding different varieties of fruit waste. The ref is aptly used here.
Ref 23 is “Antioxidants from plants protect against skin photoaging”. The ref has been used in the statement regarding the plant waste helps in fighting antioxidants. The ref is aptly used here.
Ref 29 is “Edible medicinal and non-medicinal plants: Volume 2, Springer, 2012, 1-1088, ISBN: 9789400717640, DOI: 10.1007/978-94-007-1764-0”. The problem is auto update by Mendeley (it will be cited correctly when the file is converted to word). The ref has been used in the statement as garden booster by deter in the crawling pests by plant wastes. The ref is aptly used here.
Ref 44 had an issue with Mendeley citing which has been resolved in the revised version of manuscript.
Ref 76 had an issue with Mendeley citing which has been resolved in the revised version of manuscript.
Ref 78 had an issue with Mendeley citing which has been resolved in the revised version of manuscript.
Hence, as suggested by reviewer, verification has been done in the revised version of manuscript.
Reviewer 3 Report
The paper describes the design and development of food waste inspired electrochemical platform for various applications. The paper is well organized. The literature is up to date. Features as well as downside of using the systems are very food discussed. Therefore, the paper can be accepted as submitted by the author.
Author Response
Ref: electrochem-2473850
Title: Design and Development of Food Waste Inspired Electrochemical Platform for Various Applications
- Reply to the Reviewer #3’s comments:
The paper describes the design and development of food waste inspired electrochemical platform for various applications. The paper is well organized. The literature is up to date. Features as well as downside of using the systems are very food discussed. Therefore, the paper can be accepted as submitted by the author.
Reply: Thank you for the positive feedback.
Round 2
Reviewer 1 Report
The manuscript is revised as suggested.
Author Response
Ref: electrochem-2473850
Title: Design and Development of Food Waste Inspired Electrochemical Platform for Various Applications
- Reply to the Reviewer #1’s comments:
The manuscript is revised as suggested.
Reply: Thank you for the feedback.
Reviewer 2 Report
The authors answered the questions and corrected the indicated errors accordingly.
The next sentence should be corrected:
"A non-invasive electrochemical antioxidant activity has been reported using soft carbon microelectrodes for apple skin[41] that helps in counter balancing the oxidative stress using scanning electrochemical microscopic technique."
A non-invasive electrochemical antioxidant activity measurement method has been reported using soft carbon microelectrodes for apple skin [41] that helps in counter balancing the oxidative stress using scanning electrochemical microscopic technique.
Some inaccuracies remain in the reference list: 19, 29
Please, use the correct form mentioned in the answers.
Ref. 45 and 79 was correct in the first version, but is not in the revised version.
Author Response
Ref: electrochem-2473850
Title: Design and Development of Food Waste Inspired Electrochemical Platform for Various Applications
- Reply to the Reviewer #2’s comments:
The authors answered the questions and corrected the indicated errors accordingly.
Reply: Thank you for the positive feedback.
"A non-invasive electrochemical antioxidant activity has been reported using soft carbon microelectrodes for apple skin[41] that helps in counter balancing the oxidative stress using scanning electrochemical microscopic technique."
A non-invasive electrochemical antioxidant activity measurement method has been reported using soft carbon microelectrodes for apple skin [41] that helps in counter balancing the oxidative stress using scanning electrochemical microscopic technique.
Reply: Thanks for your critical comment. We agree with your point and have suitable edited in the revised version of manuscript.
Some inaccuracies remain in the reference list: 19, 29. Please, use the correct form mentioned in the answers.
Reply: Thanks for your feedback. We agree with your point and have suitable edited in the revised version of manuscript.
Ref. 45 and 79 was correct in the first version, but is not in the revised version.
Reply: Thanks for your feedback. We have suitable incorporated in the revised version of manuscript.